# PixelGaze: Toward Pixel-Level Gaze Target Prediction in Natural Scenes

## Abstract

Following the gaze of other people and analyzing the target they are looking at can help us understand what they are thinking, and doing, and predict the actions that may follow. Existing methods for gaze following primarily focus on gaze points or heatmaps rather than objects, making it difficult to deliver clear semantics and an accurate scope of the gaze-at targets. To address this shortcoming, we propose a novel gaze target prediction method named PixelGaze, that can effectively leverage the spatial visual field of the person as guidance, enabling a progressive coarse-to-fine process for gaze target segmentation and recognition. Specifically, a prompt-based visual foundation model serves as the encoder, working in conjunction with three distinct decoding modules (e.g., FoV perception, heatmap generation, and segmentation) to form the framework for gaze target prediction. Then, with the head bounding box performed as an initial prompt, PixelGaze obtains the FoV map, heatmap, and segmentation map progressively, leading to a unified framework for multiple tasks (e.g., direction estimation, gaze target segmentation, and recognition). In particular, to facilitate this research, we construct and release a new dataset, comprising 72k images with pixel-level annotations and 270 categories of gaze targets, built upon the GazeFollow dataset. The quantitative evaluation shows that our approach achieves the mIoU of 34.9% in gaze target segmentation and 45.1% recognition accuracy. Meanwhile, our approach also achieves state-of-the-art performance on the gaze-following task.

## 1 Introduction

Even though human beings have a remarkable capability to decode the gaze behavior of others in many scenarios, realizing this task automatically remains a challenging problem (Madhusanka et al., 2022; Zhang et al., 2020a). A key step in this direction was the work by (Recasens et al., 2015), which defined the task as predicting where in an image the target person is looking. This prediction is represented as a heatmap, where the intensity at each point indicates the likelihood of it being the gaze point, with the maximum value marking the exact gaze coordinate. (Chong et al., 2020) further extended the task to handle out-of-frame gaze targets and developed methods that could track human gaze in the video. Subsequently, (Bao et al., 2022; Tu et al., 2022) use more modal information (e.g., depth, pose) to enrich the model's interpretive capacity and refine gaze prediction accuracy. Meanwhile, with the development of general segmentation models (Kirillov et al., 2023), we believe that gaze analysis systems can further improve their ability to conduct gaze target prediction.

However, due to the limitations of available datasets and the complexity of the task, pixel-level gaze target prediction in natural scenes has not yet been explored. Recently, (Tomas et al., 2021; Wang et al., 2022) made efforts to predict the target by inferring the bounding box of the gaze-at object to eliminate ambiguity. While the box is a more intuitive final output and provides an approximate object location, it still includes extraneous background details. Although limited, some studies have investigated pixel-level semantic information to conduct gaze following. For instance, the GOO dataset (Tomas et al., 2021) provides the category and mask of the gaze-at object in retail environments. (Jin et al., 2024) proposes a gaze target prediction method to identify the exact grocery item. While these methods have shown promising results in pixel-level prediction, they remain confined to a few specific objects in retail scenarios.

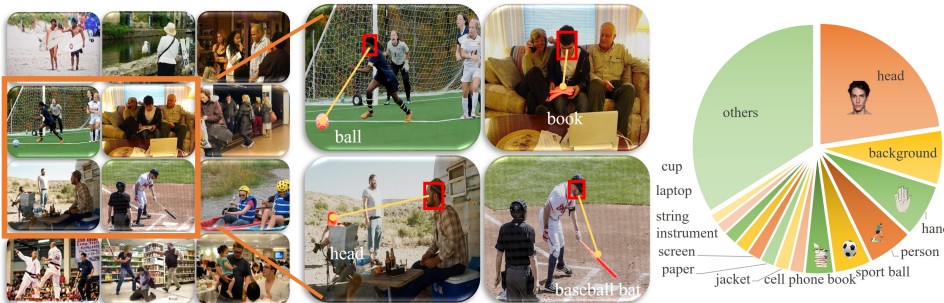

(a) Sample images from our benchmark. (b) Example annotations from our dataset. (c) Category distribution of the dataset

Figure 1: Overview of the proposed GazeSeg benchmark. (a) shows sample images from our benchmark, which includes various scenes and diverse targets. (b) shows the annotations in our dataset, including pixel-level localization and object recognition. (c) presents the distribution of the dataset, including 270 different categories.

Taking inspiration from the abovementioned challenge, we pose a critical query: How to conduct gaze target prediction in natural scenes and investigate the novel issue of segmenting the gaze-at object. This entails us creating a unified model capable of gaze following, segmenting, and classifying at the same time.

To tackle the shortage of datasets specifically designed for gaze target segmentation, we augmented the existing GazeFollow dataset (Recasens et al., 2015) with pixel-level annotations. As shown in Fig. 1, we carried out pixel-level annotation and classification of gaze targets, thereby preparing comprehensive data for gaze target prediction. It is the first pixel-level gaze target segmentation dataset for natural scenes in the third-person perspective.

Based on the dataset, we propose a unified multi-task framework named PixelGaze to achieve more precise gaze target prediction. Specifically, we design a novel progressive gaze target prediction framework with three distinct modules that conduct multiple gaze tasks (e.g., direction estimation, gaze target segmentation, and recognition) for pixel-level gaze-following. Firstly, we propose a **3D Field of View (FoV) Perception module** that uses the head bounding box coordinates as prompts and builds the corresponding 3D spatial field of view without inputting additional RGB head images. To leverage both image features and depth information (simply extracted from original images), this module generates a 3D gaze cone direction, providing a precise and reliable foundation for gaze target prediction. Secondly, we propose a **FoV-aware Heatmap Generation module** designed to predict the gaze-at locations. This module encodes the spatial FoV information, which is combined element-wise with the entire scene context in a dense prompt embedding. The integrated data is then fed into a heatmap decoder, which produces the gaze heatmap. This helps the model to locate accurate gaze following points. Finally, we propose a **Segmentation and Recognition module** to effectively leverage the mask prompt with the heatmap cues for pixel-level prediction, which obtains foreground probability masks for each position in the image. To bridge the gap between heatmap and pixel-level prediction, we adopt a differentiable numerical coordinate regression method to transform the gaze point to the mask prompt. Besides, except for task-specific losses, we propose two novel loss terms for FoV supervision and mask-heatmap matching to optimize gaze target prediction. Our contributions can be summarized as follows:

- We design a prompt-based unified framework named PixelGaze for multiple gaze-following tasks (*e.g.*, direction estimation, gaze target segmentation and recognition). This framework optimizes pixel-level prediction through a progressive localization process.

- In our solution, we propose 3D FoV perception, heatmap generation, and segmentation modules. We introduce the gaze prompt and mask prompt design, and new gaze FoV and mask-heatmap matching loss terms to improve the gaze target prediction performance.

- We extend the annotation of the Gazefollow dataset and propose PixelGaze, featuring pixel-level mask annotations of the gaze-at object across diverse natural scenes and object categories. This dataset presents novel challenges in gaze prediction research.

- We conduct extensive experiments and validate the effectiveness of our method in the gaze target prediction task, as well as to consider the benefits of pixel-level semantic information for gaze following.

## 2 RELATED WORK

**Gaze Following.** (Recasens et al., 2015) pioneered gaze following and constructed the GazeFollow dataset, which is a large-scale image dataset labeled with the locations in the image that people are looking at. Based on this, (Chong et al., 2018) further solved the out-of-frame problem by simultaneously predicting saliency maps and learning gaze angles. Meanwhile, the performance of gaze following is improved by utilizing other different auxiliary information such as body pose (Bao et al., 2022), line of sight (Lian et al., 2018; Li et al., 2019), and depth (Miao et al., 2023). In addition to detecting gaze in images, (Chong et al., 2020) proposed a new framework to understand human gaze in videos and released a video dataset called VideoAttentionTarget that contains dynamic patterns of real-world gaze behavior. Since Transformer shows excellent potential in vision tasks, (Tu et al., 2022; Tonini et al., 2023) leverages the target detection feature of the DETR architecture (Carion et al., 2020) to aid in predicting gaze position. citepwang2022gatector proposed a gaze target detection method GaTector, which utilizes an additional object detector (YOLOV4 (Bochkovskiy et al., 2020)) to identify target objects. Furthermore, (Tu et al., 2023a) proposed a unified framework to detect gaze location and gaze object bounding-boxes jointly. Although ingenious, existing gaze-following methods that rely on predicting fixed points or bounding boxes lack semantic understanding of objects, making gaze-at-objects supervision ambiguous.

**Gaze-related Datasets.** Gaze is a nonverbal cue that provides a wealth of information about people. Here, we briefly introduce the Gaze-related datasets in the visual community. For example, Gaze360 (Kellnhofer et al., 2019) and ETH-XGaze (Zhang et al., 2020b) datasets are widely used for eye' gaze estimation. However, these datasets are unsuitable for tasks involving gazing at targets. For gaze following in the third-person perspective, researchers typically use the GazeFollow (Recasens et al., 2015), VideoAttentionTarget (Chong et al., 2020), Childplay (Tafasca et al., 2023), and GOO (Tomas et al., 2021) datasets. The GazeFollow dataset includes both indoor and outdoor human activities, while the VideoAttentionTarget dataset consists of TV programs. Childplay provides a curated collection of clips with rich children's gaze information for diagnosing developmental disorders. However, these datasets lack pixel-level annotations. The GOO dataset further provides bounding boxes and pixel-level labels, but is limited to retail environments with a few objects sharing similar shapes. Existing datasets inevitably suffer from different issues for fine-grained gaze target following and recognition. To overcome this, this paper introduces a new dataset for gaze target prediction to bridge the gap between gaze information and pixel-level semantics.

## 3 METHOD: PIXELGAZE

Our goal is to automatically recognize and segment the gaze target of the designated person in a given scene and to integrate traditional gaze following tasks. Formally, given an image $x$ and the bounding box for a particular person's head $b$, the proposed PixelGaze is required to predict gaze-at instance mask and category.

**Overview**. The overview of our approach is shown in Figure 2. We solve the gaze target prediction task in a progressive manner. We adopt the mobileSAM Zhang et al. (2023) to encode the image and use the head bounding box $b$ as initial prompt, and three sequential module to obtain the gaze vector $V_g = (e_x, e_y, e_z)$, heatmap $H$, in/out of picture $c_{in}$, mask $M$, category $c$.

### 3.1 FIELD-OF-VIEW PERCEPTION MODULE

We aim to estimate a person's visible space in 3D—their field of view (FoV)—as the first step in a progressive gaze target prediction framework. This representation describes the person's potential visual interactions with their surroundings and serves as an important cue for downstream reasoning.

**Depth-Aware 3D FoV Perception.** Our approach consists of two main components: a *gaze prompt encoder* and a *lightweight gaze decoder*. Given the head bounding box $b \in \mathbb{R}^{2 \times 2}$

Figure 2: Overview of the PixelGaze framework: 1) We build a unified multi-task gaze target prediction network; 2) The progressive gaze target prediction procedure includes 3 steps: FoV perception, heatmap generation, segmentation, and recognition. 3) We adopt the lightweight design in SAM by using the prompt encoder and decoder architecture for this task.

(top-left and bottom-right coordinates), we encode its spatial location into a pair of embeddings $P_w^{init} \in \mathbb{R}^{2 \times 256}$ using positional encoding combined with learnable embeddings, following Fourier feature mappings (Tancik et al., 2020). This results in a *gaze prompt* that embeds both geometric and learnable contextual cues.

The gaze decoder receives two inputs: (1) the *image embedding* $E \in \mathbb{R}^{256 \times 64 \times 64}$ from the image encoder, providing holistic scene context; (2) a *gaze token* $Q_g^{init} \in \mathbb{R}^{1 \times 256}$, formed by concatenating the gaze prompt with a learnable embedding. A stack of cross-attention layers and MLPs enables interaction between the gaze token and image features, refining the token into a gaze-aware representation. Finally, the updated gaze token is mapped through an MLP to predict the 3D gaze vector in spherical coordinates:

$$V_g = (\vartheta, \varphi, \rho), \tag{1}$$

where $\vartheta$ is the polar angle, $\varphi$ is the azimuthal angle, and $\rho$ is the magnitude of the gaze vector.

**Person-Specific FoV Generation.** Given the predicted spherical gaze vector $V_g$, we first convert it to Cartesian coordinates:

$$V_g' = (e_x, e_y, e_z) = \begin{bmatrix} \rho \cdot \cos\vartheta \cdot \cos\varphi \\ \rho \cdot \sin\vartheta \cdot \cos\varphi \\ \rho \cdot \sin\varphi \end{bmatrix}, \tag{2}$$

where $e_z$ encodes gaze depth; omitting $e_z$ yields a 2D FoV implementation.

We then construct a spatial FoV map $I_{fov} \in \mathbb{R}^{1 \times H' \times W'}$. For each pixel $(i, j)$, let $\mathcal{M}_g^{(i,j)}$ denote the *unit direction vector* from the face center to that pixel. The FoV map is computed as:

$$I_{fov}^{(i,j)} = \begin{cases} V_g' \cdot \mathcal{M}_g^{(i,j)}, & \angle(V_g', \mathcal{M}_g^{(i,j)}) \leq \alpha, \\ 0, & \text{otherwise}, \end{cases} \tag{3}$$

where $\alpha$ is the maximum angular deviation for visibility. Pixels outside this cone are treated as blind spots. Following (Horanyi et al., 2023), we also set the values inside the head bounding box $B$ to zero to remove self-occlusion. Then, we supervise FoV generation in both spherical and Cartesian coordinate systems:

$$\mathcal{L}_{fov} = \alpha_1 \mathcal{L}_{f1} + \alpha_2 \mathcal{L}_{f2} = \alpha_1 \underbrace{|V_g - V_{gt}^{sc}|^2}_{\text{MSE in spherical coords}} + \alpha_2 \underbrace{\left(1 - \frac{V_g' \cdot V_{gt}^{cc}}{\|V_g'\|_2 \cdot \|V_{gt}^{cc}\|_2}\right)}_{\text{Angular loss in Cartesian coords}}, \tag{4}$$

where $V_{gt}^{sc}$ is the normalized spherical ground-truth gaze vector, $V_{gt}^{cc}$ is the normalized Cartesian ground-truth gaze vector, and $\alpha_1, \alpha_2$ are balancing weights. The first term enforces accuracy in gaze direction and magnitude in spherical space, while the second encourages precise orientation alignment in Cartesian space.

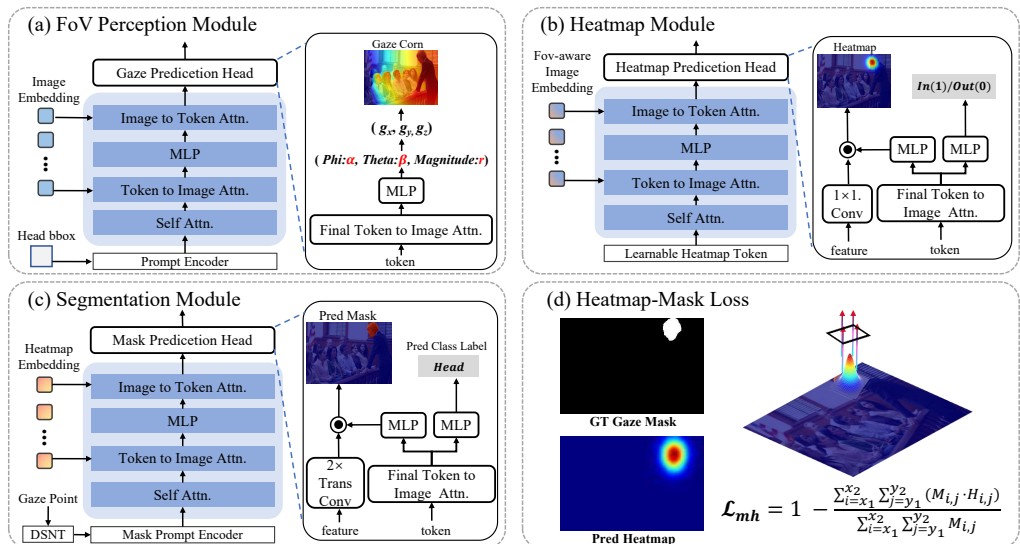

Figure 3: Illustration of the module architecture and loss function design.

## 3.2 FoV-Aware Heatmap Generation

To jointly model 3D gaze geometry and scene context, we design a *FoV-Aware Heatmap Generation Module* that predicts the coarse location of the gaze target (gaze following) before performing fine-grained segmentation and recognition. This intermediate heatmap provides a spatial prior, enabling a progressive coarse-to-fine prediction pipeline.

**FoV-aware heatmap decoder.** The spatial FoV map $I_{fov}$ is first processed by a 3D cone encoder composed of: two $2 \times 2$ convolutional layers with stride 2 (output channels 4 and 16, respectively) to match the spatial scale with the image embedding, followed by a $1 \times 1$ convolution mapping the channel dimension to 256. We then fuse FoV geometry and scene context through element-wise addition between the FoV features and the image embedding, producing *FoV-aware image embeddings* $E_{fov} \in \mathbb{R}^{256 \times 64 \times 64}$.

For the heatmap decoding, we follow the SAM lightweight decoder design and the process consists of $N_h$ transformer decoder layers (Kirillov et al., 2023). As shown in the Fig. 3 (b), the decoder takes as input FoV-aware image embedding $E_{fov}$, and a learnable *heatmap token* $Q_h^{init} \in \mathbb{R}^{2 \times 256}$. Feature interaction is realized via cross-attention between $E_{fov}$ and the heatmap token.

**Heatmap Prediction head.** After $N_h$ decoding layers, we obtain the final FoV-aware embedding $E_{fov}^{final}$ and updated heatmap token $Q_h^{final}$. $Q_h^{final}$ is passed through an MLP to predict whether the gaze target lies inside or outside the image frame. The decoded feature is processed by a Trans-Convolutional layer to output a heatmap $I_{heat} \in \mathbb{R}^{1 \times H_0 \times W_0}$.

**Loss for Gaze Following.** We supervise the I/O head for in and out of picture prediction with binary cross-entropy loss $\mathcal{L}_{io}$ and the heatmap with MSE loss. The gaze-following loss is:

$$\mathcal{L}_{gaze} = \mathcal{L}_{fov} + \beta_1 \mathcal{L}_{io} + \beta_2 \|I_{heat} - I_{heat}^{gt}\|_2^2, \tag{5}$$

where $\mathcal{L}_{fov}$ is defined, and $\beta_1, \beta_2$ are weighting factors.

## 3.3 Progressive Gaze Target Prediction

The model produced by the above method is referred to as **PixelGaze w/o Seg.** Based on this model, we further design a segmentation module and a heatmap-mask loss to achieve pixel-level prediction of the gaze target. Specifically, we conducted a two-step training process on the GazeSeg dataset. First, we trained the PixelGaze w/o Seg model using segmentation labels to produce more precise heatmap regression. Then, we froze the model and trained an additional segmentation module to achieve pixel-level prediction. Next, we will introduce each step in detail.

**Heatmap-Mask Loss**    In the first step, since the features of the heatmap are used as the input to the segmentation module, the overall performance improves when the response range of the heatmap better aligns with the segmentation labels. Therefore, before the training of the segmentation module, we designed a mask loss to constrain the range of the heatmap as shown in Fig. 3 (d):

$$\mathcal{L}_{mask} = 1 - \frac{\sum_{i=0}^{H_0} \sum_{j=0}^{W_0} (M_{gt}^{i,j} \cdot I_{heat}^{i,j})}{\sum_{i=0}^{H_0} \sum_{j=0}^{W_0} M_{gt}^{i,j}}, \tag{6}$$

where $M_{gt}$ is the ground-truth mask. Therefore, in the first step, training uses both the $\mathcal{L}_{gaze}$ and $\mathcal{L}_{mask}$ losses to jointly optimize the original PixelGaze w/o Seg model.

**Segmentation module.**    Building upon the coarse localization provided by the heatmap, we design a Segmentation Module based on a lightweight SAM-style decoder (Kirillov et al., 2023), which consists of a series of cross-attention and MLP layers. To prompt the decoder, we convert the predicted heatmap into a differentiable point representation using the DSNT layer (Nibali et al., 2018). This produces a *point prompt* that preserves spatial gradients and generalizes well to unseen layouts. The point prompt is concatenated with learnable embeddings to form the *mask token*, which is passed into the SAM decoder together with the FoV-aware image embedding from the heatmap stage.

**Pixel-Level Prediction.**    The SAM decoder outputs two predictions: (1) a segmentation mask for pixel-accurate gaze target localization, and (2) a category label for gaze target recognition. The segmentation head uses focal loss (Lin et al., 2020) and Dice loss (Milletari et al., 2016) to form $\mathcal{L}_{seg}$. The objective for gaze target prediction is:

$$\mathcal{L}_{pred} = \lambda_1 \mathcal{L}_{seg} + \lambda_2 \mathcal{L}_{cls}, \tag{7}$$

where $\lambda_1, \lambda_2$ balance the terms. We jointly optimize the complete PixelGaze framework using $\mathcal{L}_{gaze}$ (Eq. 5) and $\mathcal{L}_{pred}$ (Eq. 7), enabling a coarse-to-fine prediction pipeline from FoV geometry to pixel-level segmentation and categorical recognition. The recognition head is supervised with cross-entropy loss $\mathcal{L}_{cls}$.

## 4    EXPERIMENTS AND RESULTS

### 4.1    EXPERIMENTAL SETUP

**Datasets.** Experiments are conducted on the Gazefollow (Recasens et al., 2015) and the proposed GazeSeg benchmark to evaluate the pixel-level gaze target prediction performance. Moreover, we use the VideoAttentionTarget (Chong et al., 2020) and Childplay datasets (Tafasca et al., 2023) to test gaze following performance.

**Evaluation Metrics.** We use totally six metrics to evaluate the performance (Chong et al., 2020; Chen et al., 2023). (1) For gaze following, we adopt the commonly used **AUC**, which calculates the area under the TPR $vs.$ FPR curve. **Distance (Dist.)** denotes the L2 distance between the predicted and the ground truth coordinates of the gaze target. we examine the average distances and minimum distances when more than one annotation is available. Also, Average Precision (**AP**) is used to evaluate the performance of intra-frame and extra-frame classification in the VideoAttentionTarget datatset (Chong et al., 2020). *(2) For segmentation*, we adopt the **mIoU** and **AP50** to evaluate segmentation performance. *(3) For recognition*, we adopt **Accuracy** to measure the accuracy of whether the true category is in one of the top 1 category of its prediction.

**Implementation Details.** We train for 50 epochs on the GazeFollow for PixelGaze w/o Seg, and 10 epochs on the GazeSeg for PixelGaze model. The model is implemented in PyTorch (Paszke et al., 2019). For 3D cone, we downsample the depth map and the image feature map to a quarter of the original image size and construct the 3D cone, and empirically set the constraint angle $\alpha$ to 90°. For the image encoder and masking module setups, we adopt MobileSAM (Zhang et al., 2023) as the backbone. The loss hyperparameters are empirically set as $\alpha_1, \alpha_2 = \{1000, 100\}$, $\beta_1, \beta_2 = \{20, 10000\}$, and $\lambda_1, \lambda_2 = \{100, 10\}$. For model training, we use AdamW (Loshchilov & Hutter, 2017) optimizer with a weight decay 0.1 and an initial learning rate of 1e-4 on the PixelGaze and GazeFollow, and 5e-5 on the VideoAttentionTarget. The batch size is 16 on all datasets.

Table 1: Main comparison on the GazeFollow (Recasens et al., 2015) and VideoAttentionTarget (Chong et al., 2020). The best and the second-best results are marked in **bold** and underline.

| Method | Venue | GazeFollow | | | | VideoAttentionTarget | | | Params ↓ |
| | | Localization | | | Estimation | Localization | | | |
| | | AUC ↑ | Avg Dist. ↓ | Min Dist. ↓ | Ang° ↓ | AUC ↑ | Dist. ↓ | AP ↑ | |
|---|---|---|---|---|---|---|---|---|---|
| Random (Chong et al., 2020) | CVPR'20 | 0.504 | 0.484 | 0.391 | 69.0 | 0.505 | 0.458 | 0.621 | - |
| Fixed bias (Chong et al., 2020) | CVPR'20 | 0.674 | 0.306 | 0.219 | 48.0 | 0.728 | 0.326 | 0.624 | - |
| (Chong et al., 2018) | ECCV'18 | 0.896 | 0.187 | 0.112 | - | 0.830 | 0.193 | 0.705 | - |
| (Lian et al., 2018) | ACCV'18 | 0.906 | 0.145 | 0.081 | 17.6 | 0.837 | 0.165 | - | 55.7M |
| Chong et al. (2020) | CVPR'20 | 0.921 | 0.137 | 0.077 | - | 0.854 | 0.147 | 0.848 | 61.4M |
| Jin et al. (2021) | FG'21 | 0.919 | 0.126 | 0.076 | - | 0.881 | 0.134 | 0.880 | 60.7M |
| (Fang et al., 2021) | CVPR'21 | 0.922 | 0.124 | 0.067 | 14.9 | 0.905 | 0.108 | 0.896 | 68.8M |
| (Tonini et al., 2022) | ICMI'22 | 0.927 | 0.141 | - | - | 0.940 | 0.129 | - | - |
| (Bao et al., 2022) | CVPR'22 | 0.928 | 0.126 | - | 15.3 | 0.885 | 0.120 | 0.869 | - |
| (Gupta et al., 2022) | CVPRW'22 | 0.943 | 0.114 | 0.056 | - | 0.913 | 0.110 | 0.879 | - |
| (Jin et al., 2022) | EAAI'22 | 0.923 | 0.120 | 0.064 | 14.8 | 0.882 | 0.113 | 0.897 | - |
| (Hu et al., 2022) | TCSVT'2022 | 0.923 | 0.128 | 0.069 | - | 0.880 | 0.118 | 0.881 | - |
| (Miao et al., 2023) | WACV'23 | 0.934 | 0.123 | 0.065 | - | 0.917 | 0.109 | 0.908 | 62.0M |
| (Tu et al., 2022) | CVPR'22 | 0.917 | 0.133 | 0.069 | - | 0.904 | 0.126 | 0.854 | 43.0M |
| (Tu et al., 2023b) | TCSVT'23 | 0.921 | 0.121 | 0.068 | - | 0.931 | 0.105 | 0.914 | - |
| (Tonini et al., 2023) | ICCV'23 | 0.922 | **0.069** | **0.029** | - | 0.933 | 0.104 | **0.934** | 53.8M |
| (Tafasca et al., 2023) | ICCV'23 | 0.936 | 0.125 | 0.064 | - | 0.914 | 0.109 | 0.107 | - |
| (Tafasca et al., 2024) | CVPR'24 | 0.944 | 0.113 | 0.057 | - | - | 0.107 | 0.891 | - |
| (Song et al., 2024) | Arxiv'24 | 0.949 | 0.105 | 0.047 | - | 0.938 | 0.102 | 0.905 | 22.0M |
| (Ryan et al., 2024) | CVPR'25 | **0.958** | 0.099 | 0.041 | - | 0.937 | 0.103 | 0.903 | - |
| Human (Recasens et al., 2015) | - | 0.924 | 0.096 | 0.040 | 11.0 | 0.921 | 0.051 | 0.925 | - |
| **Ours ( w/o Seg)** | - | 0.947 | 0.110 | 0.053 | 14.0 | - | - | - | 26.5M |
| **Ours** | - | 0.953 | 0.092 | 0.043 | **10.8** | 0.943 | 0.090 | 0.930 | 30.6M |

## 4.2 MAIN RESULTS ON GAZE FOLLOWING DATASETS

To conduct comprehensive experiments for gaze following, we compare with state-of-the-art methods on both GazeFollow (image) and the VideoAttentionTarget (video) datasets. We report the latest methods' performance and model size in Table 1. Several key observations are summarized as follows: *Firstly*, 3D gaze cone construction and pixel-level semantic segmentation modules are beneficial for the task. Our method achieves superb gaze localization results, outperforming existing methods across multiple metrics, including AUC, distance, Angle, and AP. For example, our method achieves desirable results on the VideoAttentionTarget dataset in terms of AUC (0.943) and AP(0.930), demonstrating the adaptability of the proposed framework. *Secondly*, the proposed method is more efficient and accurate than existing methods. Our model builds on Mobile-SAM (Zhang et al., 2023) combined with the new design of the FoV Perception module, heatmap generation and segmentation and recognition modules. Even this, the full model efficiently accomplishes the gaze estimation task with a relatively small number of parameters, demonstrating the effectiveness. *Thirdly*, the proposed method closely approximates human performance. Table 1 reports the results of human gaze localization. It can be observed that humans still have an advantage in the distance (Dist) metric. Nevertheless, our method surpasses human observers in metrics such as AUC and AP, indicating that our approach has achieved a level comparable to humans.

## 4.3 MAIN RESULTS ON GAZESEG BENCHMARK

### 4.3.1 QUANTITATIVE ANALYSIS

Here, we focus on the pixel-level gaze target prediction. We primarily compare two categories of approaches on the PixelGaze benchmark: gaze-following methods and combine them with the mobileSAM and the CLIP model. For gaze-following methods, we threshold their heatmaps to serve as segmentation results. Moreover, we compare with a gaze object detection method (Wang et al., 2024) only applicable to retail scenes, which can predict object categories but cannot provide pixel-level segmentation results. From Table 2, we observe that using the heatmap directly as segmentation results cannot properly represent the object of interest. In contrast, our method can better highlight the object of interest and provide semantic information to pixel-level masks through prediction. To fairly compare to these methods, we utilize the gaze point from sota methods as prompt to segment the gaze target the mobileSAM model. Moreover, we use CLIP model to recognize the gaze target.

Table 2: Gaze target prediction results in terms of Localization, Segmentation and Recognition on the PixelGaze Benchmark.

| Method | Venue | Localization | | | Segmentation | | Recognition | Params ↓ |
|---|---|---|---|---|---|---|---|---|
| | | AUC↑ | Avg Dist↓ | Min Dist↓ | mIoU↑ | AP↑ | Acc↑ | |
| Gaze Following Methods | | | | | | | | |
| (Chong et al., 2020) | CVPR'20 | 0.921 | 0.137 | 0.077 | 11.9 | - | - | 61.4M |
| (Miao et al., 2023) | WACV'23 | 0.934 | 0.123 | 0.065 | 12.7 | - | - | 62.0M |
| (Song et al., 2024) | VI'24 | 0.949 | 0.105 | 0.047 | 15.6 | - | - | 22.0M |
| Gaze Following Methods + SAM + CLIP | | | | | | | | |
| (Chong et al., 2020)+SAM+CLIP | CVPR'20 | 0.921 | 0.137 | 0.077 | 23.39 | 10.62 | 33.60 | - |
| (Tonini et al., 2023)+SAM+CLIP | ICCV'23 | 0.922 | 0.069 | 0.029 | 34.35 | 13.40 | 41.26 | - |
| (Tafasca et al., 2024)+SAM+CLIP | CVPR'24 | 0.944 | 0.113 | 0.057 | 25.92 | 10.67 | 35.71 | - |
| (Song et al., 2024)+SAM+CLIP | VI'24 | 0.949 | 0.105 | 0.047 | 34.19 | 13.22 | 37.08 | - |
| (ViT-B) (Ryan et al., 2024)+SAM+CLIP | CVPR'25 | 0.956 | 0.104 | 0.045 | 33.51 | 13.38 | 37.38 | - |
| (ViT-L) (Ryan et al., 2024)+SAM+CLIP | CVPR'25 | 0.958 | 0.099 | 0.041 | 34.24 | 13.41 | 38.97 | - |
| **Ours** | - | 0.953 | 0.092 | 0.043 | **34.86** | **14.03** | **45.10** | **30.57M** |

Figure 4: Visualization of the Gaze target prediction results for the proposed method.

From Table 2 again, even with the provision of real gaze information, our method still outperforms the methods across the board in terms of segmentation ability for gaze targets. We believe that the unified implementation of gaze target prediction is a challenging task. Nevertheless, even with a small parameter count, we have successfully achieved effective localization, segmentation, and recognition of gaze targets while achieving optimal performance.

### 4.3.2 VISUALIZATION ANALYSIS

We present some quantitative results in Fig. 4. The scenarios include indoor and outdoor, and the subject include children and adults. We can observe that in the 2nd column, the method prompted by facial cues can generate the individual's specific FoV, enabling preliminary gaze analysis. Subsequently, the 3rd column illustrates the heatmap generated using the FoV prompt for prediction, which closely approximates the ground-truth heatmap shown in the 4th column. Finally, the last three columns in the figure illustrate the segmentation and recognition results. It can be observed that, based on reliable FoV perception and heatmap generation, the model can conduct gaze target prediction successfully.

Table 3: Results on the ChildPlay dataset.

| Method | Venue | AUC↑ | L2↓ | P.Head↑ |
|---|---|---|---|---|
| Gupta et al. (2022) | CVPRW'22 | 0.919 | 0.113 | 0.694 |
| Tafasca et al. (2023) | ICCV'23 | 0.935 | 0.107 | 0.663 |
| Tafasca et al. (2024) | CVPR'24 | - | 0.106 | 0.600 |
| Ryan et al. (2024)(ViT-B) | CVPR'25 | 0.949 | 0.106 | **0.715** |
| Ryan et al. (2024)(ViT-L) | CVPR'25 | 0.951 | 0.101 | 0.662 |
| Ours | - | **0.956** | **0.099** | 0.698 |

### 4.3.3 PERFORMANCE ON CHILDPLAY

We evaluated our approach in the ChildPlay dataset (Tafasca et al., 2023) to explore its potential in autism screening applications. This is an autism screening collection of carefully curated video clips of children playing and interacting with adults in uncontrolled environments. We describe

performance using the Looking At Head Precision metric (P.Head). From Table 3, in both scenarios our method outperforms existing methods, indicating its potential value.

## 4.4 ABLATION STUDIES

### 4.4.1 INFLUENCE OF THE MAIN MODULES

As shown in Table 4, when removing the total FoV perception module (w/o FoV module) and using only image embedding to execute subsequent heatmap and mask modules, the model's performance degrades on the GazeSeg dataset. Similarly, retaining the gaze module without 3D FoV construction and 3D cone encoder (w/o 3D cone, directly inputting the image embedding into heatmap decoder) faces significant performance degradation. Furthermore, we estimate the 3D gaze of the to-be-detected person directly through the global context without using the gaze prompt encoder (w/o GPrompt) to provide the corresponding head bounding box, and the model performance decreases. For constructing the FoV maps, the lack of spatial information of the depth map (w/o Depth, *i.e.*, 2D FoV) brings the performance decay too. We also conduct ablation studies by directly using the image embedding output by the image encoder (w/o Heatmap) and using a non-differentiable argmax function to obtain point cues from the heatmap instead of the DSNT layer on the segmentation phase (w/o DSTN). In the absence of either of the two above cases, the performance degrades. Overall, to achieve optimal performance, the key modules in our model are essential.

Table 4: Ablation studies of the main modules on the PixelGaze.

| Method | Segmentation mIoU↑ | Recognition Acc↑ | Localization AUC ↑ | Localization Avg Dist. ↓ |
|---|---|---|---|---|
| w/o FoV | 19.7 | 39.5 | 0.923 | 0.168 |
| w/o 3D Cone | 19.9 | 39.5 | 0.924 | 0.167 |
| w/o GPrompt | 24.2 | 41.2 | 0.939 | 0.124 |
| w/o Depth | 31.1 | 42.6 | 0.942 | 0.102 |
| w/o Heatmap | 22.5 | 38.6 | 0.947 | 0.103 |
| w/o DSTN | 32.1 | 42.9 | 0.950 | 0.093 |
| Full Model | **34.9** | **45.1** | **0.953** | **0.092** |

Table 5: Ablation studies of loss objective terms.

| $\mathcal{L}_{f1}$ | $\mathcal{L}_{f2}$ | $\mathcal{L}_{mask}$ | Segmentation IoU↑ | Recognition Acc↑ | Localization AUC ↑ | Localization Min Dist. ↓ |
|---|---|---|---|---|---|---|
| | | | 29.6 | 34.0 | 0.942 | 0.062 |
| ✓ | | | 30.6 | 37.7 | 0.944 | 0.053 |
| | ✓ | | 30.8 | 38.7 | 0.943 | 0.052 |
| ✓ | ✓ | | 30.8 | 38.8 | 0.947 | 0.053 |
| ✓ | | ✓ | 32.1 | 43.2 | 0.949 | 0.046 |
| | ✓ | ✓ | 32.3 | 43.4 | 0.951 | 0.044 |
| ✓ | ✓ | ✓ | **34.9** | **45.1** | **0.953** | **0.043** |

### 4.4.2 INFLUENCE OF THE OBJECTIVE TERMS

We also conduct ablation studies to analyze the new loss functions. Since $\mathcal{L}_{seg}, \mathcal{L}_{io}, \mathcal{L}_{cls}$ are task-specific loss terms, we primarily discuss $\mathcal{L}_{fov}$ (include $\mathcal{L}_{f1}$ and $\mathcal{L}_{f2}$ in Eq.4) and $\mathcal{L}_{mask}$. From Table 5, we find that modeling accurate gaze direction and FoV map brings significant performance improvements for localization, segmentation, and recognition. The loss of the supervised gaze angle difference in the Cartesian coordinate system brings a more stable performance improvement compared to the MSE loss in spherical coordinates. In addition, the mask loss effectively limits the prediction range to the center region of the target object, which greatly enhances the localization ability of the heatmap regression increases in Min.Dist and directly benefits the segmentation and recognition tasks.

## 5 CONCLUSION

In this paper, we present PixelGaze, a challenging benchmark for pixel-level gaze target prediction in variant scenes. The unique challenges presented by PixelGaze position it as a noteworthy benchmark in this field. We propose a novel solution, which is a unified multi-task framework for progressive pixel-level gaze segmentation and category prediction. Extensive comparative experiments and ablation studies validate that the proposed method achieves SOTA performance. We aim to encourage continued research in pixel-level gaze target prediction, with a focus on advancing the development of models that improve the performance of both segmentation and recognition. These improvements can help achieve a deeper understanding and interpretation of human behavior.

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
