# OpenReview forum: "PixelGaze: Toward Pixel-Level Gaze Target Prediction in Natural Scenes"
_ICLR.cc/2026/Conference — Submitted to ICLR 2026_

### Official Review · Reviewer_pZBG · 2025-10-29

**Soundness:** 3
**Presentation:** 2
**Contribution:** 3
**Rating:** 4
**Confidence:** 5

**Summary:**

The paper addresses gaze target prediction in natural scenes and introduces the novel task of gaze target segmentation. To enable this, the authors augment the GazeFollow dataset with pixel-level annotations, creating the first third-person, pixel-level gaze target segmentation dataset. Building on this dataset, they propose PixelGaze, a unified multi-task framework that performs gaze following, segmentation, and recognition simultaneously. The framework consists of three key modules: a) 3D Field of View (FoV) Perception that generates a 3D gaze cone using head bounding boxes and depth information to provide a precise foundation for gaze prediction.
b) FoV-aware Heatmap Generation that integrates spatial FoV and scene context to predict gaze-at locations via a heatmap.
c) Segmentation and Recognition that combines heatmap cues with mask prompts for pixel-level segmentation and recognition, bridging heatmap and mask predictions using differentiable coordinate regression.
The framework is optimized with task-specific losses as well as novel FoV supervision and mask heatmap matching losses, achieving more accurate gaze target prediction.

**Strengths:**

- The paper presents a unified framework for gaze target detection and segmentation. The segmentation task is novel in gaze behavior analysis and could be useful for providing more fine-grained detections.

- It also introduces new annotations, consisting of carefully curated segmentation masks based on samples from the GazeFollow dataset (but also see the weaknesses section)

**Weaknesses:**

1) The comparisons do not seem entirely fair. For example, in Table 1, the proposed method uses the head bounding box as a prior, whereas many of the methods listed in the table simultaneously detect head locations (e.g., Tu et al., Tonini et al.). Please confirm this and justify how this relates to the results.

2) One of the main contributions of this study is the new annotations. However, without understanding how they were collected, reviewers cannot assess whether the results are reliable. Unfortunately, there is no explanation regarding the annotations in the main paper, appendix, or supplementary video. Key details are missing, such as (a) how you annotated it, (b) how many people participated in the annotation process, (c) whether the same image was annotated by more than one person, and (d) whether additional verification of the annotations was performed. These details are crucial in gaze research, as subjectivity plays a significant role in annotating gaze images, which is reflected in the high variance of gaze point locations in the GazeFollow test set (already shown by many prior papers). It is also unclear whether the entire GazeFollow dataset was annotated, which seems unlikely given its size and the multiple gaze annotations per image, or just a portion of it. From Fig. 3c, it seems a portion. Can you clarify?

3) Not clear why there are no comparisons on the GOO dataset, which in fact supplies pixel-level annotations. Could you please explain?

4) The paper includes a “w/o DSNT” ablation, showing minor performance drops when replacing the DSNT layer with a non-differentiable argmax. However, the motivation for choosing DSNT over direct coordinate regression or simple heatmap sampling is not conceptually discussed, and the reported improvement is modest without clear justification.

5) The paper lacks a comparison with a simple yet informative baseline where the segmentation branch is replaced by a frozen semantic segmentation model (e.g., SAM) that generates a mask using the gaze point predicted by the heatmap module. Such a “naïve” setup would provide a clear reference for assessing the real contribution of the proposed segmentation and recognition branches. It would help determine whether the performance gains stem from the proposed design or could be similarly achieved by combining standard gaze prediction with an off-the-shelf segmenter.

6) It is not clear what the failure cases are or why they occur (can be given in appendix). Overall, the paper provides very little discussion or explanation of the design choices, and most of the parameters appear to have been set empirically. It also seems that no separate validation split was used for parameter tuning across the datasets. Can you explain?

7) [Minor] It seems that the Supp. video was prepared for CVPR 2025 submission, but including the name of the paper, should be changed.

**Questions:**

In addition to the above questions:

1) During progressive training, the model is first trained (PixelGaze w/o Seg) and then frozen while adding the segmentation module. The paper doesn’t explain why freezing is beneficial. For example, was joint fine-tuning unstable? It’s unclear whether end-to-end fine-tuning might improve performance. Such info can be included in an appendix, which is nowadays extensive in top conferences.

2) The loss weights $(\alpha_{1}, \alpha_{2}, \beta_{1}, \beta_{2}, \lambda_{1}, \lambda_{2})$ are said to be ``empirically set,'' but there is no description of how these values were chosen, whether they were tuned per dataset, and how sensitive the performance is to these parameters. Without this information, it is difficult to reproduce the results or assess fairness in comparison.

3) What is the significance of the segmentation and recognition tasks' results between your method and the others?

---

> ### Author Response · Authors · 2025-11-28
> **Response to Reviewer pZBG**
>
> **[W1: Head bbox information]**  Thanks for pointing out. Our design choice is motivated by the fact that the primary focus of this work is on the gaze-to-target understanding stage—specifically, how to translate a given gaze signal into a precise, semantically meaningful target representation. This allows us to isolate and advance the core challenge of where and what is being looked at, decoupled from the head detection problem. While our main results assume given head boxes, we agree that a full end-to-end evaluation is valuable. In the revised manuscript, we now clarify this setting and discuss its implications.
>
> **[W2: Detail on GazeSeg annotation]** Thanks for your question, please refer to the Response for **Reviewer KVsx Q6**.
>
> **[W3: Experiments on GOO dataset]** Thanks for your concern. We conduct experiments on GOO-syn dataset.
>
> | Method       | AUC ↑     |  Dist. ↓  |  mSoC ↑   |
> |:-------------|:-------|:-------|:-------|
> |  SamGOP[3]  | 94.7 | 0.072  | 2.70 |
> | Ours   | 96.3 | 0.069  | 2.85   |
>
> [3] Jin Y et al. Boosting gaze object prediction via pixel-level supervision from vision foundation model[C]//ECCV, 2024: 369-386.
>
> **[W4: Detail on DSNT]** Directly using argmax leads to non-differentiability in backpropagation, affecting the training effectiveness of the model. The DSNT layer computes coordinates in a differentiable manner. Specifically, DSNT normalizes the heatmap and then calculates its expected coordinates (i.e., weighted average), mapping spatial distributions (heatmaps) to numerical coordinates in a fully differentiable process. We directly used the DSNT layer to ensure that the process is differentiable. When using argmax, we need to set the gradient of this part to 1 to ensure backpropagation, but this leads to inaccurate gradient updates, affecting the training effectiveness of the model. i.e., when using argmax, its mIoU is 32.1, while using DSNT, its mIoU is 34.9, with a performance improvement of 2.8.
>
> **[W5: Baseline with SAM]**  Thanks for your suggestion. We conduct experiment using mobileSAM and CLIP as the segmentation and recognition modules, respectively. Specifically, the mobileSAM and CLIP were integrated into the PixelGaze framework without any modifications. The results are as follows.
>
> | Method       | AP50 ↑     |  mIoU ↑  | Acc ↑   |
> |:-------------|:-------|:-------|:-------|
> | Ours (w/o seg) + SAM + CLIP   | 11.96 | 32.05  | 35.53 |
>
>
> **[W6: Failure cases and parameter selection]**
> 1. **Failure cases** Thanks for your concern. In the supplementary, we provide results of prediction failures, as shown in the last three rows of Fig. A4. When the face is not visible (rows 10 to 11), the model usually cannot accurately predict 3D gaze direction, leading to errors in execution. Notably, in these failure samples, even humans find it challenging to accurately determine the gaze target.
> 2. **Parameter selection.** Thanks for your suggestions. We conducted a parameter analysis on the decoder depth as shown in the following table. Specifically, we kept the MobileSAM encoder and the segmentation decoder unchanged, and varied the depths of $N_g$ (FoV Perception Module) and $N_h$ (Heatmap Module). The results are added to the supplementary.
>
> |   \( $N_g$ \)   | \( $N_h$ \)     |  mIoU ↑  | Acc ↑   | AuC | Min Dist |
> |:-------------|:-------|:-------|:-------|:-------|:-------|
> | 6   | 4 | 28.75  | 41.9  | 0.928 | 0.051 |
> | 4   | 6 | 32.11  | 44.3  | 0.940 | 0.046 |
> | 6   | 6 | **34.86**  | **45.10** | **0.953** | **0.043** |
> | 8   | 8 | 32.21  | 44.7  | 0.941 | 0.045 |
>
> **[Q1: End to end training]** Thanks for your concern, please refer to **Reviewer KVsx W2**.
>
> **[Q2: Parameter sensitivity]** We conducted a parameter sensitivity analysis on the loss weights $\lambda_1$ and $\lambda_2$ in pixel-level prediction $\mathcal{L}_{pred}$. The results are shown in the following table.
>
> | $\lambda_1$, $\lambda_2$| mIOU | Acc | AP |
> |:-------------|:-------|:-------|:-------|
> | 50, 10 | 33.87 | 44.35 | 13.95 |
> | 100, 5 | 33.76 | 41.56 | 13.69 |
> | 100, 10 | **34.86** | **45.10** | **14.03** |
> | 100, 20 | 33.50 | 44.09 | 13.69 |
> | 150, 20 | 33.93 | 44.18 | 13.89 |
>
> When $\lambda_1$ is too small, the segmentation performance drops significantly, which in turn affects the recognition performance. Similarly, when $\lambda_2$ is too small, the recognition performance drops significantly.
>
> **[Q3: Significance of Segmentation and Recognition]**  The segmentation and recognition results demonstrate a key advantage of our approach: pixel-level semantic understanding of gaze targets, which goes beyond coarse localization used in prior methods. While existing approaches typically predict only a gaze point, heatmap, or bounding box, our method jointly produces accurate object masks and category labels, enabling not just where someone is looking, but what they are looking at—down to the object instance and shape.

---

### Official Review · Reviewer_rS8p · 2025-10-30

**Soundness:** 4
**Presentation:** 3
**Contribution:** 2
**Rating:** 2
**Confidence:** 4

**Summary:**

This paper proposes PixelGaze, a new method for gaze target prediction that goes beyond traditional gaze point or heatmap estimation. Instead of only predicting where a person is looking, PixelGaze identifies which object they are looking at and segments it at the pixel level. The method uses a coarse-to-fine pipeline guided by a visual foundation model. Starting from the person's head bounding box as a prompt, the model progressively generates a field-of-view (FoV) map, a gaze heatmap, a pixel-level segmentation map of the gaze target, along with its object category. To support this task, the authors create a new dataset of 72k images with pixel-level gaze target annotations and 270 object categories, derived from GazeFollow. Experiments show that PixelGaze achieves 34.9% mIoU for segmentation and 45.1% recognition accuracy, outperforming previous gaze-following methods.

**Strengths:**

1. This paper go further from gaze target detection to gaze target segmentation.
2. The proposed method is effective for doing this problem.
3. A dataset and benchmark GazeSeg is proposed.

**Weaknesses:**

1. No detailed description of GazeSeg. Moreover, the construction process of the benchmark is missing. The labels depend heavily on GazeFollow dataset, making the contribution limited.
2. The novelty of the proposed method is limited. The proposed method is just a concatenation of FOV perception, heatmap generation, segmentation modules.
3. The segmentation function is significantly affected by the localization performance. What is the motivation of training a new segmentation module? The three modules are trained one by one or together?

**Questions:**

1. How was the ground truth made?
2. The three modules are trained one by one or together?
3. What is the performance if use your backbone + SAM + CLIP

---

> ### Author Response · Authors · 2025-11-28
> **Response to Reviewer rS8p**
>
> **[W1: Detail and contribution about GazeSeg]** To clarify, we have provided a detailed description of the GazeSeg dataset in the **Review KVsx Q5** and in the appendix, including specific information about the dataset.
> Thank you for your comment regarding the dataset contribution. While GazeSeg is indeed built upon the GazeFollow dataset, we believe it provides meaningful and non-trivial contributions to the field for the following reasons:
> 1. **Lack of existing segmentation benchmarks:** To the best of our knowledge, there is no publicly available dataset that provides pixel-level annotations for gaze target understanding in natural, in-the-wild scenes. GazeSeg fills this gap and enables research on segmentation-based gaze target prediction—a direction previously hindered by the absence of such labels.
> 2. **Richness and diversity of GazeFollow:** Compared to other gaze datasets such as VideoAttentionTarget (VAT), GOO, or ChildPlay—which often focus on constrained settings — GazeFollow contains diverse real-world environments and a wide variety of gaze-at objects. Enhancing this already rich dataset with high-quality masks significantly increases its utility for fine-grained gaze analysis.
> 3. **Substantial annotation effort:** Generating accurate pixel-level masks for Gazefollow dataset is a labor-intensive and non-trivial endeavor. Our human-validated annotations, combined with a rigorous quality control protocol, provide a reliable foundation for future research.
>
> In summary, GazeSeg establishes the first large-scale segmentation benchmark for naturalistic gaze target understanding and opens new avenues for method development and evaluation.
>
> **[W2: Novelty of segmentation module]** The core novelty of our work lies in a progressive, modular framework that decomposes gaze analysis into sequentially refined stages (e.g., FoV perception → gaze localization → target segmentation). This design not only improves computational efficiency but also enhances interpretability and facilitates module replacement or upgrades — a flexibility lacking in most end-to-end approaches.
> In essence, PixelGaze introduces a co-designed, synergistic framework where modules interact purposefully to solve a complex, under-explored problem—pixel-level gaze target segmentation in the natural sceans.
>
> In specific, we first train a base model that includes the FoV perception module and gaze heatmap regression head—mirroring the typical setup in existing literature. Only after this stage do we add a segmentation decoder to produce pixel-level masks for the gaze target. This two-stage protocol allows us to isolate the contribution of our segmentation design while maintaining compatibility with prior work.
>
>
> **[Q1: Ground truth]** **GazeSeg** is built by extending the GazeFollow dataset with additional pixel-level masks and semantic categories. The annotation protocol for ground truth is as follows:
> 1. **Annotation pipeline**: Starting from the original gaze points in GazeFollow, we first use YOLO to detect potential target objects and then apply SAM to generate initial masks. These masks are subsequently refined and validated by human annotators using AnyLableling.
> 2. **Annotator involvement**:
> **Training set**: 15 annotators participated, with each image reviewed by **one** annotator.
> **Test set**: 15 annotators participated, with each image independently reviewed by **three** annotators to ensure reliability.
> 3. **Category selection**:
> When multiple semantic interpretations are possible (e.g., “head” vs. “person”), we prioritize **fine-grained labels** (e.g., “head” if the gaze point falls on the head; otherwise, the full “person” is annotated).
> 4. **Handling ambiguous cases**:
> In the **training set**, images with highly ambiguous or unclear gaze targets are **excluded** to reduce noise.
> In the **test set**, **all images** are annotated, including ambiguous ones, to enable comprehensive evaluation. For images with multiple gaze targets, we retain up to **three objects** in the test set (correctly segmenting any one of them is considered a success during evaluation).
>
> **[Q2: Training stages]** In our paper, we adopted two stage training include the conventional gaze analysis and gaze target prediction(segmentation decoder).
>
> **[Q3: Performance of ours + SAM +CLIP]** Thanks for your suggestion. We conduct experiment using mobileSAM and CLIP as the segmentation and recognition modules, respectively. Specifically, the mobileSAM and CLIP were integrated into the PixelGaze framework without any modifications. The results are as follows.
> | Method       | AP50 ↑     |  mIoU ↑  | Acc ↑   |
> |:-------------|:-------|:-------|:-------|
> | Ours (w/o seg) + SAM + CLIP   | 11.96 | 32.05  | 35.53 |
>
> Compared with combining the three models, our designed progressive model improves the prediction performance.

---

### Official Review · Reviewer_HKKm · 2025-11-01

**Soundness:** 3
**Presentation:** 3
**Contribution:** 4
**Rating:** 6
**Confidence:** 4

**Summary:**

In this paper, the authors explore pixel-level gaze target prediction through their proposed PixelGaze method which performs multiple tasks such as direction estimation, gaze target segmentation and recognition. PixelGaze consists of several modules dedicated to 3D FoV perception, heatmap generation and segmentation. To facilitate their multi-task method on fine-grained gaze target following and recognition, the authors also extend Gazefollow dataset by including segmentation mask ground truth annotations for the gaze-target object. The method achieves impressive results on existing gaze following datasets using traditional gaze following metrics. Additionally, they achieve SOTA on the new gaze target prediction tasks such as segmentation and recognition.

**Strengths:**

1. The extension of gaze following to pixel-level understanding that the paper is trying to solve is interesting and relevant in real-world applications.
2. The PixelGaze method has been designed well and makes sense intuitively.
3.  PixelGaze achieves good performance on the traditional gaze following task, and also on their proposed segmentation and recognition tasks.

**Weaknesses:**

1. Localization Results for Gazefollow are worse for PixelGaze, while that for  VideoAttentionTarget and ChildPlay datasets are only slightly better for some metrics (note that AP value [0.930] for PixelGaze is in bold signifying best result, but Tonnini et al. achieve higher AP value [0.934]).
2. There are several missing portions in the experimental results in Tables 1 and 2: (a) Ours (w/o Seg) for VideoAttentionTarget dataset; (b) Params column for several previous methods which have publicly available code (such as Tafasca et al. (2024), Tafasca et al. (2023), Ryan et al. (2024)). Why were these not reported?

**Questions:**

Please respond to my question in Weakness point 2. Additionally, I have a couple of questions and a comment:

1. Have you tried using mobileSAM and CLIP instead of the segmentation and recognition module, to have an apple-to-apple comparison with your baselines in Table 2?
2. What do you think are the limitations of your method?
3. Lastly, a comment: the way you have cited papers without enclosing parentheses (for instance, see lines 130-136) looks off to me. This can be rectified using a different LaTeX command.

---

> ### Author Response · Authors · 2025-11-28
> **Response to Reviewer HKKm**
>
> **[W1: Localization performance]** Thank you for your careful evaluation of our results. We agree that on the GazeFollow dataset, PixelGaze does not uniformly surpass all prior methods in every metric (e.g., AUC: 0.953 vs. the current SOTA of 0.958). However, PixelGaze achieves consistently strong results across three distinct datasets—GazeFollow, VideoAttentionTarget, and ChildPlay, demonstrating its robust generalization capability.
> Thank you for pointing out the error in the AP labeling in the table. We have corrected it in the revised manuscript.
>
> **[W2: Missing portions in the experimental results in Tables 1 and 2]** Thank you for your careful review. In the revised manuscript, we have added the results of Ours (w/o Seg) on the VideoAttentionTarget dataset, as well as the number of parameters for the compared methods with publicly available code (including Tafasca et al. 2023, Tafasca et al. 2024, and Ryan et al. 2024).
>
> VideoAttentionTarget Dataset:
>
> | Method       | AUC ↑     | Dist ↓  | AP ↑   | Params (M) |
> |:-------------|:-------|:-------|:-------|:-------|
> | Ours (w/o Seg)   | 0.926 | 0.102  | 0.790 | 26.5 |
> | Ours    | 0.943 | 0.090  | 0.930 | 30.6 |
>
>
> **[Q1: Compare to mobileSAM+CLIP]** We have tried using mobileSAM and CLIP as the segmentation and recognition modules, respectively. Specifically, the mobileSAM and CLIP were adopted without any modifications. We can observe that our designed prompt encoder and decoder improve the segmentation and recognition performance.
> | Method       | AP50 ↑     |  mIoU ↑  | Acc ↑   |
> |:-------------|:-------|:-------|:-------|
> | Ours (w/o seg) + SAM + CLIP   | 11.96 | 32.05  | 35.53 |
>
> **[Q2: Limitation]** The limitations of our method are that the segmentation task and recognition task are trained separately, which may lead to suboptimal performance. In future work, we plan to explore end-to-end training strategies to jointly optimize both tasks.
>
> **[Q3: Citation issue]** Thanks, we have revised the manuscript.

---

### Official Review · Reviewer_KVsx · 2025-11-10

**Soundness:** 2
**Presentation:** 1
**Contribution:** 2
**Rating:** 4
**Confidence:** 3

**Summary:**

The paper introduces PixelGaze, a unified model designed to predict not only where a person is looking within an image but also to produce a fine-grained, pixel-level segmentation map of the gaze target. To enable this capability, the authors also present a new  dataset, GazeSeg, an extension of the GazeFollow dataset. GazeSeg contains 72,000 images with pixel-level gaze target annotations across 270 object categories, providing a comprehensive resource for training and evaluating gaze target segmentation models in natural scenes.

**Strengths:**

GaseSeg is a dataset contribution with pixel-level masks, and semantic labels, likely to stimulate further research.

**Weaknesses:**

he novelty of the paper is somewhat limited. While the overall framework integrates multiple modules, several components are not clearly explained or insufficiently justified. The training procedure involves a two-stage setup, but it is unclear why this design is necessary or why the model cannot be trained end-to-end. As a result, the work feels more like an engineering effort that combines existing modules rather than a fundamentally novel methodological contribution. In addition, there are inconsistencies and typographical errors in the paper—for example, at lines 176 and 180, the dimension of  Pw init is described once as  R2 and once as
R1 . The description of the encoder architecture is also vague, leaving it unclear which backbone or features are used. Similarly, the section on person-specific FoV generation lacks sufficient implementation and mathematical detail to be fully reproducible.

there are also typos in related works such as line 123-124.

**Questions:**

- Why is the bounding box prediction not sufficient for this task? Can you explain more clearly why it is not semantically meaningful compared to your segmentation-based approach?
- How often is the predicted label from the bounding box actually different from the label obtained from the segmentation map? Some quantitative comparison between these two outputs would help justify the need for pixel-level segmentation.
- what is the baseline exactly for 34% in gaze target segmentation and 45% accuracy? to what baseline is this being compared to?
- in line 175 what is the encoder model?
- in table 1 where the human values are coming from -reference is missing.
- how the annotation were done, was a SAM model used for annotation? there is also a big portion of others, what is included in that, how the categories were selected for annotation?

---

> ### Author Response · Authors · 2025-11-28
> **Response to Reviewer KVsx**
>
> **[W1: Novelty and two-stage training]**  The core novelty of our work lies in a progressive, modular framework that decomposes gaze analysis into sequentially refined stages (e.g., FoV perception → gaze localization → target segmentation). This design not only improves computational efficiency but also enhances interpretability and facilitates module replacement or upgrades — a flexibility lacking in most end-to-end approaches.
>
> Meanwhile, an end-to-end training strategy yields slightly inferior performance, likely due to optimization challenges arising from simultaneously learning generic gaze features and task-specific segmentation.
>
> | Method       | AUC ↑     | Avg Dist ↓ | mIoU ↑  | Acc ↑   |
> |:-------------|:-------|:---------|:-------|:-------|
> | End-to-end   | 0.949 | 0.099  | 34.41 | 43.86 |
> | Two-stage    | **0.953** | **0.092**  | **34.86** | **45.10** |
>
>
> **[W2: Clarification on $P_w^{init}$ and $Q_w^{init}$]**
> We need to clarift that the $P_w^{init}$ and $Q_w^{init}$ is not same dimension, because the output for $P_w^{init}$ is used to represent the head bounding box, while the output for $Q_w^{init}$ is used to represent the gaze token.
>
>
> **[Q1: Why not bounding box]** The bounding box can provide a coarse localization of the gaze target, which may include irrelevant background areas. In contrast, our segmentation method can accurately delineate the shape and extent of the gaze target, providing more precise and semantically meaningful information about what the person is looking at.
>
> **[Q2: Mask for label prediction]** To analyze the impact of the segmentation mask on object recognition, we conducted an ablation study as shown in the table below. The results show that using only the bounding box information to predict object categories achieves an accuracy of 43.4%, which is lower than our mask-based approach, demonstrating the effectiveness of the segmentation mask in recognizing gaze targets.
> | Method       | Acc ↑   |
> |:-------------|:-------|
> | Ours without using mask | 43.4 |
> | **Ours**  | **45.1** |
>
> We have illustrated the comparison between bounding boxes and segmentation maps in Figure 1 and 4. Specifically, gaze heatmaps are often blurry, and bounding boxes cannot accurately represent the shape and size of the gaze target. In contrast, segmentation maps can precisely depict the contours and details of the gaze target.
>
> **[Q3: 34% baseline]** The baseline is that removes our designed prompt encoder and decoder for segmentation and recognition, directly using moblieSAM and CLIP as the segmentation and recognition modules.
>
> **[Q4: encoder model in line 175]** The 175 line refers to the **Gaze Prompt Encoder**.
>
> **[Q5: Missing reference]** Thanks for your correction. We referred to the work of Adria et al. (2015) [1], and have added the relevant citation in the manuscript.
> [1] Recasens A, Khosla A, Vondrick C, et al. Where are they looking?[J]. Advances in neural information processing systems, 2015, 28.
>
> **[Q6: Annotation process]** Thank you for your questions regarding the annotation process.   **GazeSeg** is built by extending the **GazeFollow** dataset with additional pixel-level masks and semantic categories. The annotation protocol is as follows:
> 1. **Annotation pipeline**: Starting from the original gaze points in GazeFollow, we first use **YOLO** to detect potential target objects and then apply **SAM** to generate initial masks. These masks are subsequently refined and validated by human annotators using **AnyLableling** [2].
> 2. **Annotator involvement**:
> **Training set**: 15 annotators participated, with each image reviewed by **one** annotator.
> **Test set**: 15 annotators participated, with each image independently reviewed by **three** annotators to ensure reliability.
> 1. **Category selection**:
> When multiple semantic interpretations are possible (e.g., “head” vs. “person”), we prioritize **fine-grained labels** (e.g., “head” if the gaze point falls on the head; otherwise, the full “person” is annotated).
> 1. **Handling ambiguous cases**:
> In the **training set**, images with highly ambiguous or unclear gaze targets are **excluded** to reduce noise.
> In the **test set**, **all images** are annotated, including ambiguous ones, to enable comprehensive evaluation. For images with multiple gaze targets, we retain up to **three objects** in the test set (correctly segmenting any one of them is considered a success during evaluation).
> [2] https://github.com/vietanhdev/anylabeling

---

### Meta-Review · Area_Chair_4GYs · 2026-01-05

**Summary:**

The paper presents a method and a new dataset for segmenting and recognizing attended objects in images. The paper received mixed initial reviews. While reviewers found the task interesting and recognized the contribution of the new dataset (GazeSeg), they raised several concerns. Common issues included: (1) limited novelty of the proposed method; (2) insufficient justification of the method design; (3) missing details regarding the new dataset; and (4) limited experimental validation, including missing baselines, ablations, and benchmarks. The author rebuttal has address some but not all of these concerns. In particularly, the novelty of the proposed method is not well addressed.

Given the remaining concerns and the overall lukewarm reviewer evaluations, the AC cannot recommend acceptance in its current form. The authors are encouraged to further incorporate reviewer feedback, clarify the conceptual contributions, and strengthen the presentation and validation of the work, and to consider resubmitting to a future venue.

**Reviewer Concerns:**

The authors provided a rebuttal and a revised version of the paper. The revision includes additional experimental results and more dataset details, which help address some of the concerns, including design justification (partially if not all), dataset description, and experimental validation. However, the concern regarding limited novelty remains largely unaddressed.

The AC notes that the novelty concern primarily relates to the conceptual contribution. While the authors clarified aspects of the technical implementation in the rebuttal, the broader conceptual innovation remains unclear. Prior work has explored detecting and segmenting gaze targets (e.g., GazeFollow and GOO), sequential refinement strategies have also been studied (e.g., Jin et al., ECCV 2024), and prompt-based segmentation has been popularized by SAM variants. It is therefore not evident how the proposed design offers new conceptual insights for this task, or generalizable principles beyond this specific task.

Additionally, several important elements, such as dataset details, design rationale, and key experiments, are added in the revision and appendix. These components are expected to be clearly presented in the initial submission.

**Reviewer Scores:**

The rebuttal has addressed some of the reviewers' concerns. The AC expects that Reviewers HKKm and pZBG, who are less concerned about the novelty of the proposed work, may consider increasing their scores. However, it remains unclear whether such changes would be sufficient to bring the paper above the acceptance threshold.

---

### Decision · Program_Chairs · 2026-01-26

Reject